# Association between Synonymous SNPs of *SOX10* and Plumage Color and Reproductive Traits of Ducks

**DOI:** 10.3390/ani12233345

**Published:** 2022-11-29

**Authors:** Teysar Adi Sarjana, Gongyan Zhang

**Affiliations:** 1Key Laboratory of Agricultural Animal Genetics, Breeding and Reproduction, College of Animal Science and Technology, Huazhong Agricultural University, Wuhan 430070, China; 2Faculty of Animal and Agricultural Science, Diponegoro University, Semarang 50275, Indonesia

**Keywords:** single-nucleotide polymorphisms, gene mutation, gene sequencing, egg traits, age at sexual maturity, egg weight

## Abstract

**Simple Summary:**

A total of six novel SNPs were found from 11 identified SNPs in the duck *SOX10* gene. Three of them were associated with reproductive traits, whereas two were associated with plumage color.

**Abstract:**

Mutations in the *SOX10* gene affect the plumage color of chickens and pigeons. The mutation also causes abnormal pigmentation of the skin and hair color, as well as postnatal growth retardation and reproduction problems in humans and mice. In this study, we investigated the association between the *SOX10* gene and plumage color and reproductive traits of ducks using SNPs. We found six novel SNPs from 11 identified SNP sites using direct sequencing for PCR products from three different mixed DNA pools. We found two coding SNPs to be associated with the plumage color of ducks (ZJU1.0 Chr1. g.54065419C>T and g.54070844C>T), and found three coding SNPs associated with the reproductive traits of ducks (g.54065419C>T, g.54070844C>T, and g.54070904C>T), which were age at sexual maturity, body weight at sexual maturity, and the Haugh unit for egg quality traits and egg production in different productive periods. These results also indicated that the T alleles of the three SNPs of the coding region of *SOX10* contribute to lower reproductive traits.

## 1. Introduction

Pigmentation in ducks is a substantial issue for down feather commodities. Although color changes do not necessarily affect quality, the pigmentation of poultry products can be critical to visual acceptance and consumer preferences. Plumage color has become an important consideration in the feather and down industries in China, the USA, and some other countries [1,2]. China is the country that produces the most live ducks and duck meat. Therefore, the production of down and feathers from ducks substantially contributes to the country’s national income. About 90% of down production in China comes from ducks, and the remaining 10% comes from geese, with a total export value of more than USD 248 million in the first quarter of 2015 [3,4]. White down is generally preferred and is more valuable than that produced from dark feathers, as white down is invisible when applied to light-colored coverings, especially in the apparel (jackets, vests, boots, etc.) and bedding industries [5]. Moreover, white down has become an important material for testing the regulations of the International Down and Feather Bureau (IDFB) [6].

A white plumage strain for ducks was reported to be obtained by crossing white *Kaiya* and white *Liancheng* [7]. This study revealed that, instead of white plumage, 80% of offspring have gray plumage, and the color is variable, with a white belt running from the neck to the chest and with colored plumage on the heads, wings, backs, and tails. Black plumage, a new phenotype, was observed in the F_2_ population when the offspring were intercrossed. The gene that controls variations in plumage color in these populations has not been identified.

Mutations in *SOX10* affect plumage color in chickens [8] and pigeons [9], and cause abnormal pigmentation of skin and hair in humans [10,11] and mice [12,13]. *SOX10* is a major regulator of neural crest formation and is involved in the specification of the fate of neural crest progenitors to form the pigment cell lineage [14]. This gene is essential for stem cell maintenance and plays additional roles during tissue homeostasis and regeneration in adults [15,16]. *SOX10* is a member of the *SOXE* family transcription factor, which has a well-established role in melanocyte biology and is essential for melanocyte migration and survival [17]. Aoki et al. reported that defective melanocyte development occurred in *SOX10*-deficient mice, and *SOX10*-injected mice embryos displayed a massive increase in the number of pigment cells (Trp-2-expressing cells) [14]. Related to reproductive traits, *SOX10* expression is negatively associated with the expression levels of estrogen and progesterone hormone receptors [18]. In some cases, mutations of *SOX10* may be linked with Kallmann syndrome and hypogonadotropic hypogonadism, which delay spontaneous puberty [19]. *SOX10* is expressed in both sexes during normal mouse development, and its protein is able to activate the transcriptional targets of *SOX9*, which may be involved in sex reversal [20]. Mertelmeyer et al. reported that, in female mice, *SOX*10 is important for both prenatal-hormone-independent and pubertal-hormone-dependent branching of the mammary epithelium, as well as for proper alveologenesis during pregnancy [21]. Because these *SOX10*-related functions have been reported in other species, whether *SOX10* mutations are associated with plumage color or reproductive traits in ducks, which has not yet been reported, is worthy of investigation.

In this study, we aimed to identify SNPs in the *SOX10* gene and their association with variations in the plumage color and reproductive traits of crossbred ducks. We first screened the SNPs in the *SOX10* gene. Then, we used three of the screened SNPs as markers to perform further association analysis. The results of this study can be used to crossbreed ducks to obtain white down, which is important in the feather and down industries.

## 2. Materials and Methods

### 2.1. Animals and Samples

We derived an F_2_ population by crossing white *Kaiya* and white *Liancheng*, as described in a previous study [7]. We evaluated a total of 899 F_2_ ducks of both sexes for their plumage color and further haplotype studies, and we observed 643 female ducks for their reproductive traits. For each individual, we collected blood samples from the brachial wing veins of the ducks from three plumage color groups: white, black, and gray (Figure 1), and we stored the samples at −20 °C. Our procedures for animal husbandry and the collection of blood samples followed the guidelines of The Tab of Animal Experimental Ethical Inspection of the Laboratory Animal Center, Huazhong Agriculture University, No. HZAUDU-2018-001. We extracted genomic DNA using the phenol–chloroform extraction method, which we stored at 4 °C until further use.

### 2.2. SOX10 Gene Sequencing and Identification of Polymorphisms

Polymerase chain reactions were performed using five pairs of designed primers according to the *SOX10* gene sequence (Anas platyrhynchos NC_051772.1 GeneID: 101803890), as shown in Table 1. All SNPs that were involved in this study are located on chromosome 1 of assembly ZJU1.0. Our aim was to amplify and sequence the coding region of the *SOX10* gene.

For efficiently identifying individual variations, we performed the initial screening using Sanger sequencing with the DNA pools, followed by individual genotyping, which we used to verify the results from the initial screening. We constructed three DNA pools, each consisting of 20 individuals with a shared plumage color, i.e., one pool was white, one pool was gray, and one pool was black. We adjusted the concentration of each individual DNA to 120 ± 20 ng/μL before making the pools, and we transferred the same volume of DNA from each individual into the pools. We prepared the three DNA pools for faster SNP screening by sequencing the PCR products. We used the DNA Star SeqManII sequence analysis software, NCBI’s online BLAST software, and Clustal Omega (EMBL-EBI 2015) to visualize and analyze the sequence data. We used SIB’s Expasy translate tool, an online bioinformatics resource portal, to translate the nucleotides into protein sequences using SWISS-MODEL. Among the identified SNPs, we then selected 3 SNPs (ZJU1.0 Chr1. g.54065419C>T, g.54070844C>T, and g.54070904C>T) to genotype the individuals in the population using polymerase chain reaction–restriction fragment length polymorphism (PCR–RFLP) analysis.

### 2.3. Trait Records

The traits that we recorded in this study were plumage color and reproductive traits. Plumage color consisted of three groups: white, grey, and black. Egg traits included shell strength, egg shape index, Haugh unit (HU), and yolk color index, whereas reproductive traits included age at sexual maturity (ASM), body weight at sexual maturity (BWSM), egg weight at sexual maturity (EWSM), egg weight at peak production (EWPP), and cumulative egg production at day 120 (EP120), 150 (EP150), 180 (EP180), 210 (EP210), 240 (EP240), 270 (EP270), 300 (EP300), 330 (EP330), 360 (EP360), 390 (EP390), and 420 (EP420). HU is a measurement of egg albumen quality based on the height of the egg white and is calculated by the following formula: 100 log [10 H − 1.7w^0.37^ + 7.6] [22].

### 2.4. Statistics

Based on individual genotyping, we calculated minor allele frequency (MAF) and heterozygosity for the three SNPs. We performed a statistical test to determine if the Hardy–Weinberg Equilibrium (HWE) existed in our population of ducks using PLINK v1.07 [23]. We then used PLINK output files to create the linkage disequilibrium (LD) block, estimated using Haploview 4.2 [24]. We used SPSS 19.0 (SPSS Inc. USA) to analyze associations and correlations between genetic variables, while Bonferroni correction was applied to adjust the significance level when testing multiple SNPs simultaneously. All ducks were fed the same diet and managed in the same manner. We included no fixed effect in the analyses, because we used only female ducks for the quantitative traits.

## 3. Results

### 3.1. Segregation of Plumage Color and Reproductive Traits in F_2_ Ducks

By observing the 899 F_2_ individuals, we categorized three plumage color groups, including 282, 399, and 218 individuals with white, grey, and black plumage, respectively. The summary statistics for each reproductive trait can be found in Appendix A Table A1.

### 3.2. SNP Identification, Linkage Disequilibrium, and Haplotypes

We identified eleven mutation sites on the *SOX10* gene from the initial screening with PCR and pooled sequencing (Table 2 and Figure 2), six of which were novel SNPs. We did not test all of these SNPs in the entire population. We selected only g.54065419C>T, g.54070844C>T, and g.54070904C>T, because RFLP analysis can only be applied to these three SNPs, which were necessary for the following experiments.

The results of linkage disequilibrium (LD) revealed three LD blocks between g.54065419C>T, g.54070844C>T, and g.54070904C>T in the *SOX10* gene (Figure 3). Haplotype analysis of duck reproductive traits from the combination of the genotypes of three SNPs confirms a significant effect on the later phase of egg production at 270–420 days of age (Table 3). Such effects may be due to at least one of the SNPs that affecting the phenotypes (Appendix A Table A1), or the combined effects of multiple SNP within a haplotype. Individuals with H4 showing lowest persistency of egg production, while individuals with H2 showing highest persistency of egg production. Among these haplotypes, individuals with H5 (43.10%) had the highest frequency. The significant haplotype association pattern happened when the cumulative egg productions were recorded longer than 240 days.

### 3.3. Genotype Association of Duck Plumage Color

The genotypic frequencies of the three examined SNPs for different plumage colors are presented in Table 4. These SNPs were in the Hardy–Weinberg equilibrium (expected genotypic frequencies are: 0.12:0.45:0.43; 0.02:0.26:0.72; 0.06:0.38:0.56 for each SNP, respectively, *p* > 0.05). According to the genotyping and association analysis, we found significant associations between mutations and duck plumage color. Two of the three SNPs were associated with plumage colors (*p* < 0.01 for g.54065419C>T and g.54070844C>T, whereas we found no significant association for g.54070904 C>T, with *p* > 0.05).

The genotype frequency of the C/T of g.54065419 was higher in ducks with grey plumage, whereas the C/C frequency of g.54070844 was higher in ducks with white plumage. These association results suggested that the relationship of the three examined SNPs with plumage pigment intensity is not independent, and any mutation in this particular site may lead to variations in duck plumage pigment intensity. For the two SNPs, the presence of the C allele, whether in homozygous or heterozygous genotypes, might lighten the pigment intensity of the plumage to grey or white. These results support the idea that *SOX10* mutations lead to melanin pigment reductions [8,25].

### 3.4. Association of Duck Reproductive Traits

All three SNPs that we examined in this study were associated with duck reproductive traits (Appendix A Table A1). Mutation g.54065419C>T was associated with ASM, EP150, EP390, and EP420. g.54070904C>T was significantly associated with EP300 and EP360–EP420, and g.54070844C>T was significantly associated with EP360–EP420. All three SNPs showed strong associations (*p* < 0.003) with at least one of the duck egg production traits, especially when the cumulative egg numbers were recorded for longer time periods (EP390 and EP420). Generally, the T allele of each SNP was associated with lower reproductive traits. This indicated that a delayed start to egg production might not be compensated by the increased duration of egg production.

## 4. Discussion

In this study, the frequency of the heterozygous genotype (C/T) of g.54065419C>T was highest in ducks with gray plumage, which is in agreement with the results of our previous study [7]. This emphasizes that this heterozygous genotype may relate to duck plumage color differences, especially for gray plumage. Domyan et al. stated that epistatic autosomal mutations of the *SOX10* transcription factor gene considerably decrease the regulation of the target genes of Tyrp1 in the feathers of recessive red birds [9]. It also substantially decreases tyrosinase synthesis and changes the plumage to reddish colors, and masks and reduces the expression of melanin biosynthesis. Epistatic genes can mask each other’s presence or combine to produce a duck with grey plumage. The heterozygous state may not completely reduce plumage pigmentation due to incomplete dominance of duck *SOX10* mutation. Although mutations in the *SOX10* gene of humans and mice (in the heterozygous state) lead to a dominant-negative effect by disrupting the functions of the wild-type SOX10 protein, it has a milder effect on the Waardenburg–Shah (WS4) phenotype than that seen in homozygotes due to the incomplete blocking of *SOX10* functions [26,27]. WS4 syndrome is characterized by enteric ganglionitis and pigmentation defects [13].

In this study, we proposed that the mutations on duck *SOX10* have pleiotropic effects affecting plumage color, cumulative egg production, and age at sexual maturity. The ducks were reared with ad libitum feeding program. Sexual maturity with an ad libitum feeding program should be reached by 16 weeks or 112 days of age [28]. Therefore, ducks with the T/T genotype for g.54065419C>T in *SOX10* reached sexual maturity three weeks later at an average age of 135 days. Our results also suggested that the delayed start in egg production might be unable to be compensated during later stages of egg production. These are consistent with those of Cherry and Morris [28] and Wright et al. [29], who suggested that with delays in sexual maturity, sexual development and the capacity for further growth appear to be lost, which eventually leads to a decrease in total egg production. In female individuals, reduced SOX10 levels impair mammary gland function. The role of SOX10 in epithelial branching morphogenesis are not restricted to the prenatal phase, but is equally relevant to the second phase of expansion during puberty. SOX10 might interact with the pathways that control this expansion, including the estrogen, progesterone, growth hormone, and EGF receptor pathways [21]. Thus, in this study, SOX10 might have been responsible for the expression of different reproductive traits, especially in the late phase of production.

SOX10 has been identified as a driver of melanoma progression, a cancer that develops from melanocytes which are neural crest derivatives. Reproductive disorder diseases, such as basal-like breast carcinoma (BLBC), is also linked with *SOX10* gene mutations. Similar to other breast cancer problems, BLBC shows symptoms of a lack of expression of estrogen receptors and progesterone receptors [30]. Estrogen receptors (ERs) modulate reproductive biological activities, such as reproductive organ development and bone modeling [31]. From here, *SOX10* gene action can be linked with sexual development and skeletal or bone modeling. In the present study, ASM was associated with the SOX10 genotype.

Drummond and Fuller demonstrated that mice with the *αERKO* gene (αER knock out) fails to differentiate into the follicle and seemed incapable of ovulation. The mice with *βERKO* gene also shows a reduction in the oocyte number, ovarian dysregulation, and an increasing number of unruptured follicles [32]. Ivanov et al. reported that the gene expression of Estrogen Receptor 1 (*ER1*) is negatively correlated with *SOX10* [30]. The *FOXA1* gene, which opposes *SOX10* expression, cooperates with *ER1* to maintain luminal identity in breast cancer, whereas *GATA3*, *XBP1,* and *CA12* showing a negative correlation with *SOX10*, also cooperate in *ER1* signaling. Thus, *ER1* and *FOXA1* activities are essential for the function of *SOX10*. González-Morán revealed that *ER1* (also known as *ER-α* in chickens) is differentially expressed in both chicken ovaries during development. Both the left and right chicken ovaries can respond to estrogen, which directly acts through their interaction with ER-α [33]. The differential expression of ER-α is involved in asymmetric ovarian development. Reductions in the number of oocytes and less-developed ovaries are why a more intense genotype effect occurs in the later stages of egg production, which affects the total egg production in later periods. Therefore, in this study, we calculated egg production based on the cumulative number of eggs that were laid until a particular observation period.

We found a significant positive correlation (*p* ≤ 0.01) between age at first egg and egg weight (r = 0.35). This r value is larger than that reported by Cankaya et al. [34] regarding the contribution of age at sexual maturity to chicken reproductive traits (r = 0.21). In other words, we found strongly negative but favorable correlations between age at sexual maturity and egg production at 120 (−0.64), 180 (−0.78), 210 (−0.69), 240 (−0.61), 270 (−0.55), and 300 (−0.50) days (Appendix A Table A2). Since a correlation coefficient of larger than 0.5 is considered to indicate a large effect [35]. However, why the T/T genotype is associated with a lower HU value remains unclear. Since we performed the HU analysis in this study using eggs collected during peak production, while we found a positive significant correlation between EWSM and EWPP (*p* ≤ 0.01), one possible explanation could be that egg weights at the same age while observing HU values are relatively heavier.

Mutation positions of all synonymous SNPs, Arg67Arg, His162His, and Gly182Gly, are on the *SOX10* coding region. His162His lies on the conserved region of the HMG box, but the other two are not in the range of this region (Figure 4). Because all the SNPs that we found are synonymous, we identified no amino acid sequence substitutions. However, as discussed above, phenotypic effects of those SNPs were observed. Roulin and Ducrest reviewed a number of association analyses related to the *MC1R* gene and found that synonymous SNPs share the same total number of mutations compared to nonsynonymous SNPs and have a similar proportional average of nonsignificant associations per species (5.3 and 5.5, respectively) [25]. However, they only mentioned that an average of 2.2 nonsynonymous mutations per species is associated with coloration, without mentioning the average numbers of synonymous SNPs associated with coloration. Another review of associations of human diseases revealed that, from over 2000 disease cases, both synonymous and nonsynonymous SNPs shared a similar likelihood and affected the size of the association [36].

In several occurrences, synonymous SNPs are associated with a particular disease or trait. Codons changing from frequent codons to rare codons in a cluster of infrequently used codons affecting cotranslational folding timing may result in altered function [37]. Silent SNPs (synonymous) have shown the ability to affect mRNA splicing and stability. Synonymous SNPs might affect protein translation kinetics and protein folding activity [38]. Rare codons, which refer to synonymous SNPs, appear to influence the translation rate, causing a protein-folding effect, with the third base in the codon having the largest effect [36]. In the present study, we were unable to trace the effect of synonymous SNPs, as mentioned above, but the synonymous mutations of g.54065419C>T, g.54070844C>T, and g.54070904C>T, all occurring in the third base codon, might have manifested the effect. This follows from an association study by Wang et al. [39], who reported three synonymous SNPs related to white plumage in geese. Thus, we speculated that the observed synonymous SNPs in the *SOX10* gene of ducks might be responsible for duck plumage pigmentation and reproductive trait variation. An example of third base changes in synonymous SNPs is CGT>CGG (arginine), which was demonstrated by Sauna and Kimchi-Sarfaty [36], having a change in the relative synonymous codon usage (RSCU) value from 0.48 to 1.21 (ΔRSCU 0.73). A positive ΔRSCU might be associated with acceleration changes in local translation elongation rates.

## 5. Conclusions

We found six novel SNP from 11 identified SNP sites in the *SOX10* gene. This study shows that mutations in the *SOX10* gene affected duck plumage color and reproductive traits, as two out of the three observed SNPs were associated with duck plumage color variations, and all three of these SNPs were associated with reproductive traits. All the genotype variations of the observed effects of the three SNPs were more intense at older production ages, with ducks with the T/T genotype having the lowest reproductive traits. This is the first report stating that synonymous mutations of the *SOX10* gene are associated with duck reproductive traits.

## Figures and Tables

**Figure 1 animals-12-03345-f001:**
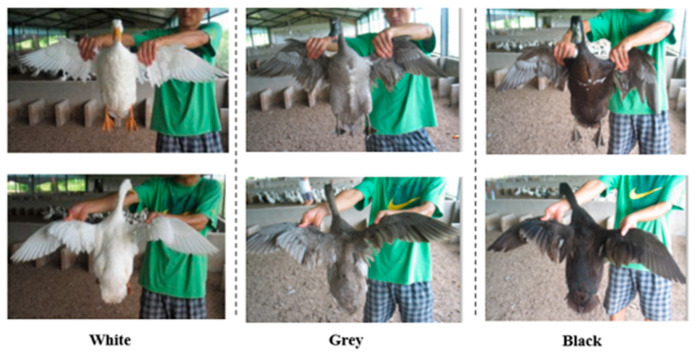
Three groups of phenotypes of duck plumage color.

**Figure 2 animals-12-03345-f002:**
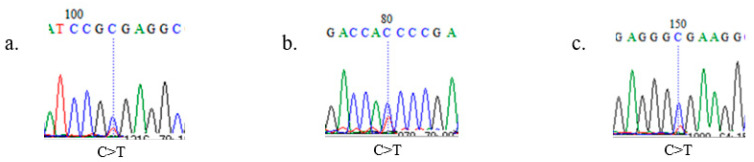
Sequence results showing SOX10 single-nucleotide polymorphisms: (**a**) g.54065419; (**b**) g.54070844; (**c**) g.54070904.

**Figure 3 animals-12-03345-f003:**
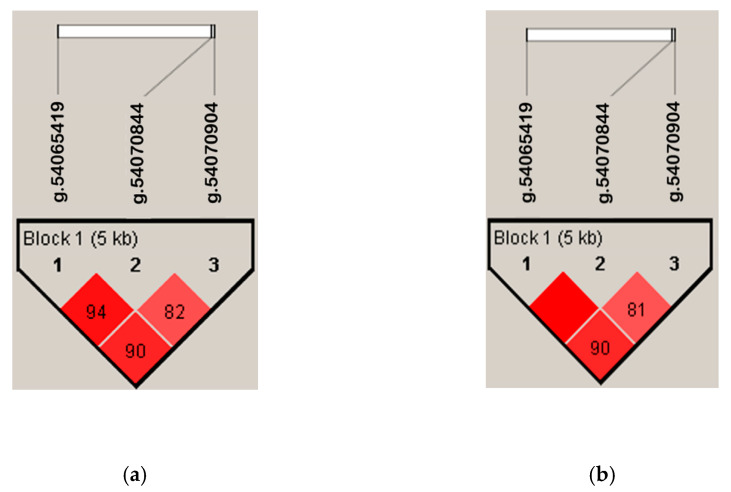
LD blocks of duck plumage color (**a**) and duck reproductive traits (**b**), consisting of g.54065419C>T, g.54070844C>T, and g.54070904C>T.

**Figure 4 animals-12-03345-f004:**
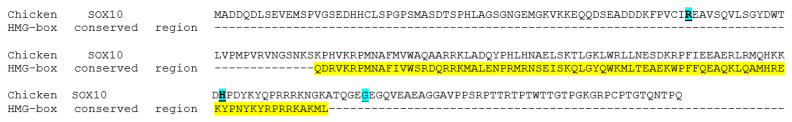
Protein BLAST of *SOX10* gene by the SWISS-MODEL. Positions of g.54065419C>T (Arg67Arg), g.54070844C>T (His162His), and g.54070904C>T (Gly182Gly) are highlighted in blue.

**Table 1 animals-12-03345-t001:** Primers designed to amplify the *SOX10* gene sequence.

Primer Name	Primer Sequence (5′ to 3′)	Product Size (bp)	Tm (°C)
S10-E1-F	ATTAGTAAAAACCAAGCCTC	242	53
S10-E1-R	CTTGTTACTTCCATTGACCC		
S10-E2-F	ACCACCACTGCCTCTCGCC	225	62
S10-E2-R	TCTTGTTACTTCCATTGACCCG		
S10-E3-F1	TTTCTCACACACCTGCCC	330	59
S10-E3-R1	TCACCTGAGGAGTGTTCTG		
S10-E3-F2	GCTGCTGAACGAAAGCGACA	289	63
S10-E3-R2	GGTTCACAAAGACCCAGGACTC		
S10-E4-F	CTCCAAAGCCCAGGTGAA	245	59
S10-E4-R	ATGGCAGTGTAAAGAGGACG		

**Table 2 animals-12-03345-t002:** Identified single-nucleotide polymorphisms (SNPs) in duck *SOX10* gene.

SOX10 SNP	Mutation	Location	Amino Acid Change
g. 54065311	G>A	Exon 2	Ser31Ser
g. 54065419	C>T	Exon 2	Arg67Arg
g. 54070844	C>T	Exon 3	His162His
g. 54070853	C>T	Exon 3	Tyr165Tyr
g. 54070904 *	C>T	Exon 3	Gly162Gly
g. 54070940 *	C>T	Exon 3	Gly191Gly
g. 54071020 *	C>T	Exon 3	Pro195Leu
g. 54071499 *	A>C	Exon 3	Gln222Lys
g. 54071948	C>T	Exon 4	Pro248Leu
g. 54072026 *	G>T	Exon 4	Leu398Val
g. 54072059 *	A>C	Exon 4	Pro423Thr

***,** novel SNP in *SOX10* gene.

**Table 3 animals-12-03345-t003:** Haplotype analysis of duck plumage color and reproductive traits.

Parameter	Means of Haplotypes of Reproductive Traits and Egg Quality [g. 54065419 | g. 54070844 | g. 54070904]					*p*-Value
	CTT (H1)	CCT (H2)	CTC (H3)	TCC (H4)	CCC (H5)	
Haplotype Count (Frequency)	157 (0.1221)	21 (0.0162)	150 (0.1163)	391 (0.3043)	554 (0.4310)	
Age at sexual maturity	121.38 ± 1.79	116.50 ± 3.61	120.58 ± 1.71	121.49 ± 0.86	121.36 ± 0.64	0.93
Egg weight at sexual maturity	48.54 ± 1.11	50.50 ± 2.25	45.32 ± 0.89	46.25 ± 0.57	46.8 ± 0.41	0.18
Egg production at 150 days	22.08 ± 1.22	26.83 ± 2.66	21.87 ± 1.24	22.12 ± 0.70	22.66 ± 0.48	0.80
Egg production at 240 days	99.24 ± 2.32	105.17 ± 7.81	100.20 ± 2.05	99.73 ± 1.22	102.66 ± 0.75	0.18
Egg production at 270 days	124.68 ± 2.70 ^ab^	130.83 ± 4.56 ^a^	125.63 ± 2.23 ^ab^	123.91 ± 1.42 ^b^	128.35 ± 0.82 ^ab^	0.03
Egg production at 300 days	150.42 ± 2.7 ^AB^	158.33 ± 4.75 ^A^	152.87 ± 2.36 ^AB^	149.01 ± 1.67 ^B^	155.12 ± 0.90 ^AB^	0.01
Egg production at 330 days	175.99 ± 3.00 ^AB^	182.50 ± 7.94 ^A^	180.26 ± 2.52 ^AB^	174.69 ± 1.91 ^B^	181.57 ± 1.01 ^AB^	0.01
Egg production at 360 days	201.77 ± 3.35 ^AB^	210.33 ± 8.95 ^A^	207.24 ± 2.69 ^AB^	199.49 ± 2.17 ^B^	206.72 ± 1.18 ^AB^	0.01
Egg production at 390 days	225.84 ± 3.69 ^ab^	239.00 ± 8.97 ^a^	232.60 ± 2.89 ^ab^	223.45 ± 2.47 ^b^	231.36 ± 1.35 ^ab^	0.02
Egg production at 420 days	248.28 ± 4.08 ^ab^	263.67 ± 8.67 ^a^	256.41 ± 3.22 ^ab^	247.19 ± 2.77 ^b^	255.05 ± 1.54 ^ab^	0.02
Haugh unit (HU)	76.86 ± 1.78	78.12 ± 2.28	79.34 ± 1.50	80.90 ± 0.86	80.94 ± 0.56	0.17

Values with different superscript in the same row differ significantly (*p* < 0.05), values with different uppercase superscript are extremely significantly different (*p* < 0.01).

**Table 4 animals-12-03345-t004:** Association between duck plumage color phenotypes and *SOX10* genotypes.

SNPs	Genotype	Genotypic Frequency			MAF	*χ* ^2^	*p*-Value
		White	Grey	Black			
	C/C	0.12	0.14	0.15			
g.54065419	C/T	0.16	0.26	0.08	0.34	50.58	0.000 **
	T/T	0.03	0.05	0.02			
	C/C	0.30	0.27	0.16			
g.54070844	C/T	0.08	0.09	0.10	0.15	18.88	0.001 **
	T/T	0.003	0.004	0.004			
	C/C	0.18	0.24	0.12			
g.54070904	C/T	0.12	0.18	0.11	0.25	5.17	0.083
	T/T	0.01	0.02	0.02			

** *p* ≤ 0.01, highly significant association. MAF: minor allele frequency.

## Data Availability

Not applicable.

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
