# Peer review of "Association between Synonymous SNPs of SOX10 and Plumage Color and Reproductive Traits of Ducks"

_animals, 2022, doi:10.3390/ani12233345_

Round 1

Reviewer 1 Report

There are many English grammar mistakes which makes it difficult to evaluate the manuscript. I did not fix these as I think the authors should first send the manuscript for editing. 

I cannot evaluate this study without an ethics clearance number. I could not find this in the manuscript.

Author Response

Dear editors / reviewer

  1. The English polishing service was applied to this manuscript by a professional agency.
  2. Research ethics is provided as attached document. And corresponding information was added in lines 84-87 in the manuscript.

Reviewer 2 Report

This is a straightforward candidate gene association analysis on duck plumage color and its reproductive traits. Using F2 cross based on white Kaiya and white Liancheng as well as moderate size of sample population, the associations were done both on SNP-level and haplotype-level.

There are a few comments that the authors should clarify:

1. I suggest the authors include a supplementary table that presents the summary statistics of the tested phenotypes. Also, one question regarding this is as the two founder lines were both “white ducks” based on their breed names, what is proportion of ducks with “grey” and “black” colors in your population.

2. For the identified SNPs in SOX10, it is advised to add the corresponding frequencies in Table 2.

3. In your association analysis, were there any fixed effects that you had accounted for, such as sex?

4. Have any of your association results been found in another duck population before? It is suggested to include such references in your discussion.

Author Response

Dear reviewer:

  1. Sentences were added in line 136-138 regarding to the summary statistics and the numbers of ducks with different plumage color.
  2. We do not have the complete data for all the frequencies, since the 11 SNPs were initial screened by PCR and pooled sequencing. The explanation is also added in line 140-144.
  3. Sentence was added to address this issue, in line 131-132.
  4. Best to our knowledge, there were no available related reference before discussing correlation between reproductive traits with SOX 10 mutation.

Reviewer 3 Report

In their paper entitled, “Association between Synonymous SNPs of SOX10 and Duck Plumage Color and Reproductive Traits”, Sarjana and Yazhang explore the role of SNPs within Sox10. They study several elements of how Sox9 affects feathering and reproductive characteristics. This topic is of interest to the feather industry but also of interest to developmental biologists.

I would like the authors to address the following concerns:

1. The NCBI lists NW_004676738.1 as being obsolete. Please update the sequence to the current version.

2. The chromosomal location of the SNPs should be listed in Table 1.

3. Percentages of haplotypes are shown but you also need to show how many birds were analyzed for each characteristic shown in the paper. Was this performed for the 20 birds representing each color? Please clarify.

4. Define Haugh unit at first use to make this easier for those not in the field to understand.

5. On lines 92-94, you indicate that you pool DNA from 20 ducks with similar color and then perform PCR and sequencing. How do you know that all 20 ducks for each color contain this sequence? You might have just amplified the sequencer from a single ducks DNA within the pool.

6. You indicate the amino acid which is changed by the SNPs but it would be helpful to refer to the full amino acid sequence shown in Fig. 3 at this point in the paper. Similarly, it would be helpful if you also refer to the SNPs in the figure legend of Fig. 3.

7. In lines 280-283 In the literature, you suggest that some codons for particular amino acids might slow the assembly of the protein. Could these effects effect particular steps in Sox10 function. Are there any indications of the specific functions of the Sox10 regions containing the identified SNPs? For instance, in Gallus gallus I see that amino acids 56-96 are needed for dimerization, while amino acids 154-192 are for DNA binding. Could changes in translation or folding at these sites offer any significant insights? 

8. Lines 145-147 – “All three SNPs were associated with plumage color variants (P<0.01 for g.287084C>T and g.292560C>T, whereas there was no significant association for g.292620 C>T with P>0.05).” 

This sentence is confusing and inaccurate.  Please rephrase it to state “SNPs g.287084C>T and g.292560C>T were associated with plumage color variants (P<0.01), whereas g.292620 C>T showed a similar trend but the association was not statistically significant (P>0.05).”

9. For the haplotype study, where are H1 – H5 located within the Sox10 sequences. Can you identify the specific relationships with the SNPs you identified. In some instances, the haplotypes seem to show no relationship with the identified SNPs. Please explain these data more clearly and show their relationship to regions within the Sox10 gene.

10. Lines 152-153 – “The presence of allele C might lighten pigment intensity into grey or white plumage color.” 

But if this were true, you would expect the CC genotype would have the highest genotypic frequency in the white group. Your data shows this for SNP g.292560 but not for g.287084 nor g.292620. Please explain how you came to this concept.

11. On line 182, please define WS4.

Author Response

  1. The duck SOX10 sequence was updated into NC_051772.1 based on the reference assembly ZJU1.0, in line 96. And all the SNP positions were BLASTed and updated to the reference assembly ZJU1.0.
  2. Table 1 was modified or remove the position information. And the chromosome information is added in lines 18 and 110.
  3. The number of individuals for haplotype study is the same with that for association analysis on plumage color, which is 899. The number of 20 for each color category were only used for the primary step to detect variation. After the existences of those variations were confirmed, then 899 samples was used for association studies. Revisions and improvements made to the lines 80-81 and table 3.
  4. Revisions and improvements for HU has been made to the line 122-123.
  5. Revisions and improvements made to the line 101-113. In brief, the DNA pools were constructed by equal amount of DNA, to make sure that each individual had the equal chance to be amplified; the pooled sequencing results (Figure 2) shows that both alleles were amplified for each SNP; the following association studies using single individuals also confirms that the pool sequencing result is reliable.
  6. Sequence variation of SNP and the figure legend has been revised in line 180.
  7. Yet we still cannot find this fact but In female individual recently reported that reduced Sox10 levels impair mammary gland function. The role of Sox10 in epithelial branching morphogenesis is not restricted to the prenatal phase, but equally relevant to the second phase of expansion during puberty. Sox10 might interacts with the pathways that control this expansion including the oestrogen, progesterone, growth hormone and EGF receptor pathways thus in this research might responsible for different reproductive traits expression especially in the late phase of production. Revisions and improvement has been made to the line 247-252.
  8. Revisions and improvements made to the line 186-187.
  9. Revisions and improvements made to the line 158-160.
  10. Revisions and improvements made to the line 192-194.
  11. WS4 refer to Waardenburg-Shah (WS4) syndrome which is characterized by enteric aganglionosis and pigmentation defects. Revisions and improvements made to the line number 224-226.

Round 2

Reviewer 1 Report

I need clarity on the method section.

- Why was the individual samples pooled? 

- Which sequencing platform was used? If Sanger sequencing was used then any individual variation within each pool will be lost as only the sequence from one fragment will be visible? 

- If there is individual variation how did you determine the SNPs?

- Indicate the reference sequence that was used to identify the SNPs.

- How did you select the coding SNPs? or were these the only SNPs located in the coding region?

- "All examined SNPs with minor allele frequency (MAF) and heterozygosity more than 0.05 were then selected and genotyped" This is unclear. How did you determine a MAF of the pooled samples or are you referring to the genotyping of the larger population? If so where did you get the MAF from then?

- Why was PCR-RFLP chosen? It is a relatively old technique. qPCR or NGS would have given more information and then all of the SNPs can be included. 

Author Response

- Why was the individual samples pooled? 

Because pool sequencing as the initial screen of variation, is a cost and labor efficient method. The detected SNPs were genotyped individually, the results verified the pool sequencing method. Revisions and improvements made to the lines 101-103.

- Which sequencing platform was used? If Sanger sequencing was used then any individual variation within each pool will be lost as only the sequence from one fragment will be visible? 

Yes, Sanger sequencing was used. We do detected variation through pooled Sanger sequencing, and we used Peakpick2 software to calculate the inferred allelic frequencies. Then the real allelic frequencies were detected by individual genotyping. The maximal differences between inferred and real allelic frequencies are less than 10%, mostly are about 5% (data not shown). Thus, for those detected SNPs, there is no obvious lost information from pooled Sanger sequencing, so it is less likely that there is other missing variation. Also, during the pool construction, we applied additional attention for making sure that each individual had the equal chance to be amplified. Related information can be found in lines 105-107.

- If there is individual variation how did you determine the SNPs?

SNPs were initially determined by pooled Sanger sequencing. As shown in Figure 2, in the Sanger sequencing results, any site with obviously double peak is a possible SNP. Then these possible SNPs were verified by individual genotyping.

- Indicate the reference sequence that was used to identify the SNPs.

- How did you select the coding SNPs? or were these the only SNPs located in the coding region?

For the initial screening of SNPs, primers were designed for only amplifying the coding region of SOX10. So these were the only SNPs located in the coding region. Revisions and improvements made to the line 98. In line 113, “coding” was deleted since all identified SNPs were coding.

- "All examined SNPs with minor allele frequency (MAF) and heterozygosity more than 0.05 were then selected and genotyped" This is unclear. How did you determine a MAF of the pooled samples or are you referring to the genotyping of the larger population? If so where did you get the MAF from then?

MAF and heterozygosity were calculated based on individual genotyping results, not pooled samples. Revisions and improvements made to the lines 128-129.

- Why was PCR-RFLP chosen? It is a relatively old technique. qPCR or NGS would have given more information and then all of the SNPs can be included. 

PCR-RFLP is chosen for its accuracy and cost efficiency. For SNP genotyping, KASP is one option (which is qPCR based), it is high throughput, but it less accurate than RFLP, usually 2 technical repetitions were needed. Melting curve-based qPCR for genotyping is another option which has less accuracy and thus needs at least 3 technical repetitions. RFLP is relative accurate because the restriction enzyme recognizes and cut the DNA sequence in a strict way. NGS is relative accurate if the read depth is high enough, while the cost increases along with the read depth, which is much less cost efficient than PCR-RFLP.